# Unleashing Knowledge Sharing in Emerging Economy Startups: A Multilevel Analysis

Roberta Dutra de Andrade [1,*], Paulo Gonçalves Pinheiro [1], Matheus Dantas Madeira Pontes [1] and Thayanne Lima Duarte Pontes [2]

1   Business and Economics Department, NECE-UBI—Research Center for Business Sciences, University of Beira Interior, 6201-001 Covilhã, Portugal; pgp@ubi.pt (P.G.P.); matheus.pontes@ubi.pt (M.D.M.P.)
2   Business and Economics Department, UNIFOR University of Fortaleza, Fortaleza 60811-905, Brazil; thayannelima@unifor.br
*   Correspondence: roberta.andrade@ubi.pt

**Abstract:** The knowledge-sharing process in digital startups is under development in current discussions, even though its importance for sustainable economic growth is acknowledged. This paper analyses the connections and traits regarding how knowledge is distinguished and shared at different levels in an emerging economy. Twelve focus groups were conducted with 72 startup founders, managers, and employees, and in-depth interviews reveal that, in contrast to the results from studies about more extensive firms, individuals consider knowledge sharing based on their particular judgment of the absorptive capacity of the recipient and the perceived gains associated with the maturation of individual and organisational image and reputation. Digital cultural factors inherent in this type of enterprise, remote first, meritocracy, and online community participation, have directly influenced the adoption of digital knowledge-sharing systems. Individuals desire to share knowledge for recognition, to establish credibility, and to establish connections with investors and mentors. A communal and collaborative atmosphere can foster the exchange of information among employees, influencing the intention to share knowledge. Knowledge sharing is reinforced when employees perceive knowledge sharers as specialists. Incentives and intra-organisational reward campaigns, talent improvement programs, external training sessions, workshops, and collaborative team assessments can cultivate personal relationships. A theoretical framework has been proposed that can examine digital startups' effectiveness on micro-level elements. In emerging economies, social rewards are personally more critical than financial achievements. Our empirical statements reinforce the arguments that the digital age, the pandemic, and the migration crisis have substantially changed most aspects of knowledge sharing.

**Keywords:** knowledge sharing; knowledge acquisition; knowledge exchange; startups; emerging economy

## 1. Introduction

Organisations' knowledge is a crucial resource in the race to achieve a sustainable competitive advantage in a dynamic economy [1]. Researchers have recently studied knowledge creation and sharing between members, teams, and other organisations [2–4]. A possible explanation for the growing interest is that knowledge flow creation and control are essential to positively impact cost reduction, shorter time spent in the development and conclusion of new product projects, team performance, innovation capabilities, and overall organisation performance [5,6].

Knowledge seen as a valuable resource, which we are concerned with exploring through startups, can be understood with several interpretations. For Nonaka [7], it is a dynamic process to justify personal truth beliefs. For Ipe [8], it is the interaction of messages

(information) with the views and commitments of its holders according to their perspectives, intentions, or specific postures that translate into actions for the creation, acquisition and sharing of knowledge. Wang and Noe [9] present an organisational view of knowledge, understanding it as the positive use of data and information that incorporates the values and contextual experiences [10] that become vital for enterprise sustainability and the generation of competitive advantages [3,11]. According to Burns ([12], p. 14), "Small firms are not just scaled-down versions of large ones." Notably, in emerging economies, startups emerge as potential promoters of development as they explore innovative activities in the market using digital technologies, where access to information can be a crucial factor for survival. In their early stages, they rely on something other than significant capital investments; therefore, the sharing and transfer of information have been adopted as instruments for creating and developing new knowledge that can be distributed among the startup's components, impacting innovation and the performance of organisations [13,14].

Despite their growing importance, studies on digital startups remain under-theorised [15–17]. Most current quantitative approaches in consolidated economies neglect how cultural factors can influence the individual perception of the benefits of sharing knowledge to promote innovation in organisations, resulting in a general sustainable competitive advantage [11,18–20]. By studying new niches in emerging markets, our study sheds light on informal regulatory practices for acquiring and sharing knowledge in multiplatform companies with a solid online presence and how digital entrepreneurial skills influence social network interactions to increase organisational performance. A few studies have included startups, most of which used quantitative approaches in advanced economies [11,20,21]. Virtual network infrastructures and digital technologies support knowledge transmission and create new opportunities and challenges for entrepreneurs to cooperate with external partners by sharing knowledge to access resources and build sustainable competitive advantages [22,23]. This study results from research on knowledge management in startups and describes how knowledge is created and how it flows between persons, groups, and firms. The main objective is to analyse the connections and traits concerning the establishment, assets, and knowledge sharing in startups from different industries and with varying degrees of maturity, considering their antecedents, motivations, and perceived results.

This study aims to answer the following research questions: (i) Why do individuals consider sharing knowledge? (ii) How can organisational culture influence the intention to share and use knowledge management systems? (iii) How do individuals' personalities interact with situational factors, and how is knowledge sharing affected? (iv) How can personal relationships with peers, teams, and other organisations be developed based on this sharing?

This research is justified by the relevance of the theme, confirmed by the growth of scientific production and the lack of similar qualitative studies on startups from emerging economies, reflecting their uniqueness and highlighting the constructions of sharing. Moreover, this research also contributes to shedding light on startups from various industries and economies and with varying maturity levels from their founders', managers', and employees' perspectives.

## 2. Literature Review

### 2.1. Knowledge Creation and Sharing through Startups

As newly established and high-tech companies, startups play a crucial role as primary drivers of organisational innovation by identifying the demand for product development in lucrative target markets [2,5,14,24]. Although different in nature or objectives, these organisations emerge with the similar purpose of aggressively growing their businesses in markets understood as highly scalable [25]. Recognised as an innate entrepreneurial ecosystem component, startups now assume a critical function in driving the advancement of emerging markets. These entities are necessary for emerging economies to be subject to

a possible lower exploration of opportunities in new market niches and a lower attraction of investors interested in developing these organisations [26].

Knowledge plays a vital role in an organisation's continued survival and sustainable competitive advantage, and this perception becomes more relevant in digital organisations such as startups. These companies are more knowledge intensive [27], and the most successful startups maintain their competitive edge by effectively accessing, generating, managing, and leveraging practical knowledge [28]. Nonaka and Takeuchi [7] classify knowledge as explicit or implicit. Explicit knowledge can be extensively accessible, measurable, and acquired through formal education. Tacit or implicit knowledge, on the other hand, concerns what we can perform but are not necessarily able to verbalise to others; it is difficult to measure, as beliefs and experiences shape it, and it can be acquired via informal training [29]. All knowledge is originally tacit and inherent in individuals [30]. Both can convert into the other through socialisation, externalisation, combination, and internalisation [7]. Knowledge can be created and shared by peers, teams, and organisations through group learning to create and acquire novel knowledge. Social interactions hold greater significance in these contexts than cognitive processes [31].

Knowledge sharing and transfer often need clarification [32]. Alavi and Leidner [33] equate and define them; they can be seen as the widespread dissemination of knowledge across the entire organisation. Wang and Noe [9] distinguish the meaning of knowledge sharing as disseminating knowledge from the holders to others. In contrast, knowledge transfer encompasses sharing knowledge from the holders to receivers who acquire and apply it. Knowledge transfer typically describes the movement of knowledge among individuals, teams, and organisations, facilitating the exchange of knowledge [34]. Knowledge exchange includes transmitting or sharing knowledge and seeking knowledge from individuals [9]. Aligned with the definition of Bock and Kim [35], the terms sharing and transfer and exchange of knowledge are considered interchangeable and interpreted as an individual's attitude towards sharing and actively seeking knowledge from others, inside and outside the organisation, to obtain personal and organisational competitive advantages.

On this path, new entrepreneurs usually seek partners with knowledge and expertise to assist them in addressing the "institutional gaps" related to administrative issues such as incipient market infrastructure, legal/regulatory systems, and financial and market development [36]. To foster the growth and sustainability of these ventures, universities, incubators, and accelerators are intensifying their efforts to support entrepreneurs in achieving desired outcomes of a more extraordinary entrepreneurial mindset while seeking to link them with more varied partners [37,38]. The literature aligns the utilisation of newly acquired information to address pivotal factors in startup development. Implementing policies to handle this knowledge effectively necessitates ensuring organisational responsiveness and learning [39]. Consequently, investigating knowledge management (KM) in startups aims to adopt innovation by cultivating new knowledge through adopting cooperative and novel technologies [40–42].

Knowledge is created through research and development, training, and drawing on organisational history [43]. Since research and formal training are expensive and startups typically need more resources [44], experiential learning is central to the new knowledge-creation process [45]. Informality is abundantly present [46] to facilitate the creation and sharing of knowledge; direct interaction and in-person discussions among staff are employed to enhance knowledge exchange [47], which can be guided by ethics and emotional intelligence [48].

Nonaka [7] elucidated how individuals and organisations can attain knowledge through diverse formal and informal mechanisms, internally encompassing know-how or experience-based knowledge and externally involving know-what or task-based knowledge, vital components of knowledge that can be shared and leveraged. Internal knowledge acquisition involves sharing knowledge between individuals and different units within a company. On the other hand, external knowledge acquisition pertains to knowledge

sharing among a company and its external partnerships, which could encompass other businesses and individuals.

Once knowledge is internally or externally acquired, it must be propagated by persons within groups or across ventures. Tacit knowledge held by individuals serves as the raw material for an upper stage of expertise to generate additional benefits [49]. Additionally, shared knowledge with an organisation and its teams enriches active knowledge depositories and creates new solutions [50]. Furthermore, this shared knowledge contributes to the development of human capacity and has a profound impact on the long-term performance of firms, enabling them to build sustainable competitive advantages [45].

The management approach based on shared value and socioeconomic benefits has grown in recent decades [51]. As a result, organisations work to create shared value with their teams and communities to ensure long-term sustainable growth [52]. The theories of exchange and social capital reasoned action, social networks, and structural holes have been employed to gain a more profound insight into information flow and the impact of relationships within networks on knowledge sharing. The theory of reasoned action focuses on the consequences of different behaviours, including sharing knowledge, mindset, and personal rules. The theory of exchange and social capital delves into operational, interpersonal, and cognitive aspects related to knowledge sharing. Lastly, the theory of structural holes and networks examines how the relationships among individuals, both within and outside organisations, impact the network's connections and how factors such as network size, connectedness, and perceived paybacks can either clog gaps or create them in the information flow within the network.

### 2.2. Theory of Structural Holes and Networks

Complex and multifaceted, the concept of social networks is a trend frequently investigated to describe the interconnectedness among individuals, inside and outside organisations, and the drift of information to knowledge exchange. The analysis of relationships in networks involves three significant elements, structural [53], interpersonal [54], and cognitive [55], as already explained in a previous section. Each dimension is itself a composite of many variables. Dey [34] and Zhao [35] are pioneers in researching structural holes and reliable and fragile interpersonal links in discussing networks' performance as links in sharing knowledge and resources to obtain and maintain competitive advantages.

Drawing from the theory of structural holes, recognised as a practical manifestation of social capital, numerous researchers suggest that an organisation's network size is a significant determinant of innovation. The level of connectedness, or its absence, among partners within the network can create more advantages and opportunities [56,57]. As gaps in organisational information flow, structural holes indicate that the links at each end can access different information flows, change their displacement and direction, and maintain the possibility of obtaining unconnected partners [58]. Moreover, structural holes can also bring other benefits concerning the volume, innovation, and quality of information circulating in the network of more connected individuals and organisations [59]. This leads to the first research question:

**RQ1:** *Why do individuals consider sharing knowledge?*

### 2.3. Grounded Action Theory

Based on grounded action theory, personnel assess the potential outcomes of various behaviours' attitudes towards knowledge sharing [60]. Additionally, several scholars propose that attitudes, subjective norms, the effectiveness of communication channels, and absorptive capacity are predictors that influence an individual's intention to share knowledge [57,61].

A blend of external incentives, a sense of mutual benefit in the relationship with managers, a feeling of personal value, and the organisational climate are indicative factors in the theory of action that promote and encourage knowledge sharing [60]. However,

despite demonstrating which positive attitudes can contribute to sharing information, the studies found the need to propose a best practices manual. Moreover, they need to be more conclusive in establishing how to improve the behaviours already implemented in organisations. So far, no investigations that have been applied to this context have found startups that showed high relevance, leading to the second research question:

**RQ2:** *How can organisational culture influence the intention to share and use knowledge management systems?*

### 2.4. Social Exchange and Share Capital Theory

The social exchange theory is a predictive framework for awareness of knowledge-sharing behaviours among individuals within an organisation [62,63]. Previous studies investigating knowledge transfer within organisations have identified emotions such as gratefulness, reliance, personal commitment, and legitimacy as critical conditions influencing knowledge transfer [63,64]. Research has indicated that leaders who foster pioneering thinking and allow employees to share information management generate a favourable organisational environment characterised by non-judgmental attitudes, leading to a sense of fairness and trust and promoting commitment [65].

Several authors (e.g., [63,66,67]) suggest that knowledge transfer within teams primarily involves components of social capital, encompassing operational, interpersonal, and cognitive elements. The operational component pertains to the models of ties and impacts on information sharing among individuals. The interpersonal component focuses on interactions among organisational stakeholders to understand the relationships between actors. The cognitive component relates to the resources that contribute to understandings, illustrations, and shared systems of social and cultural norms [63,66], putting forward the third and fourth research questions:

**RQ3:** *How do individuals' personalities interact with situational factors, and how is knowledge sharing affected?*

**RQ4:** *Based on this sharing, how can personal relationships with peers, teams, and other organisations be developed?*

### 3. Methodology
*Research Methods*

This research is a qualitative study [68] categorised as exploratory-descriptive [69]. A multiple case study was implemented [70] as an inquiry approach and was explored throughout 2022. Sample selection, purposeful and theoretical [71], as pictured in Table 1, followed several criteria: (i) distribution of the sample at different stages of maturity, from recent market entry to internationalisation, so that one might verify whether or not there were similarities in the manner in which knowledge is managed within the organisation, whether in its creation, acquisition, or sharing; (ii) all should have participated in incubation or acceleration programs and undertaken some effort of knowledge acquisition and sharing among persons, both inside and outside the organization; (iii) startups should use advanced digital technologies to offer innovative products or services, use multisized platforms, and have a robust online presence. The objects of study were 12 startups of distinct industries, originating from emerging countries, with products already validated and in commercialisation.

**Table 1.** Characterisation of the sample.

| Interviewed | Industry | Operating Time | Number of Employees | Types of Interviewees | Maturity | Incubation/ Acceleration | Investment Support |
|---|---|---|---|---|---|---|---|
| Startup A | EduTech | 6 years | 60 | founder & directors | operates in the national market | National | bootstrap & capital venture |
| Startup B | SoftTech | 3 years | 10 | founder & directors | operates in the regional market | National | bootstrap & capital venture |
| Startup C | HardTech | 2 years | 5 | all team | beginning to selling | regional | bootstrap |
| Startup D | HardTech | 6 years | 15 | founder & directors | internationalised | regional, national & international | bootstrap & economic subsidy |
| Startup E | SoftTech | 4 years | 8 | all team | operates in the national market | National | bootstrap & economic subsidy |
| Startup F | HardTech | 2 years | 6 | all team | operates in the regional market | regional | bootstrap & capital venture |
| Startup G | Fintech | 7 years | 38 | founder & directors | internationalised | International | bootstrap & capital venture |
| Startup H | Construtech | 5 years | 6 | all team | internationalised | national & international | bootstrap & capital venture |
| Startup I | Agrotech | 2 years | 9 | all team | beginning to selling | National | bootstrap & capital venture |
| Startup J | SoftTech | 4 years | 47 | founder & directors | operates in the national market | Nacional | bootstrap & economic subsidy |
| Startup K | Agrotech | 1.5 years | 12 | all team | beginning to selling | Nacional | bootstrap & economic subsidy |
| Startup L | Fintech | 8 years | 69 | founder & directors | internationalised | national & international | joint venture |

For data collection, semi-structured in-depth focus group interviews [71] were conducted with the directors, founders, and team members of the startups, as this method is widely used for market analysis and can serve as a basis for future survey research. The 72 interviewees were split into 12 focus groups [72] to perceive information and feelings that individuals have about certain specific issues and map, through data triangulation [73], the institutional practices carried out and their perceptions about type, nature, and extent of sharing and motivations for sharing knowledge, besides seeking to detect whether the developed methods interfered with the companies' performance [72].

It is worth clarifying that the focus groups comprised individuals from the same organisation. Therefore, we present the same amount of startups and focus groups. Another six startups were contacted but were unable or unwilling to participate in the research. The pre-test interview to validate the research instrument was discarded after adjustments were made, and a new startup at the same maturity stage was interviewed in substitution to avoid information bias and other forms of bias. The homogeneity of the group and the interaction between participants were ensured to allow good data accuracy. The interviews within the new focus group were closed when the data saturation criterion was reached, which occurs when the researcher can anticipate what will be said in the next group due to similar responses. Finally, it is emphasised that the results achieved through focus groups are naturally not generalisable.

Content analysis was conducted using predefined categories derived from the knowledge management current literature, social networks, and structural holes. This approach aimed to delve into the essence of meanings and identify thematic groups for further analysis [74]. Almost 22 h of interviews were transcribed and analysed into NVivo software, version 13, and the main speeches were grouped according to the categorisation tree (nodes, attributes, and relationships) shown in Table 2.

**Table 2.** Content analysis categorisation tree.

| Moderating Variables | | | |
|---|---|---|---|
| **Moderating Variables** | incubation/acceleration | | |
| | maturity level | | |
| | organisational size | | |
| | industry type | | |
| | company type | | |
| **Sharing Type** | know how | | managerial knowledge |
| | | | improve competitiveness |
| | | | create innovation |
| | know that | | play routines |
| | | | improve qualification |
| **Sharing Nature** | tacit | | |
| | explicit | | |
| **Sharing Extension** | between individuals | | |
| | intra-organizational | | |
| | inter-organizational | | performance |
| **Sharing Motivation** | externals | environmental factors | organisational context |
| | | | interpersonal & team features |
| | | | cultural features |
| | internals | individual factors | |
| | | motivational factors | |
| | | behavioural factors | |

## 4. Results

Considering knowledge management throughout startups, they all had several means of gathering, preserving, and disseminating explicit knowledge. Survivors or early-stage startups utilised unpaid cloud storage space platforms that provided unrestricted access to all employees. As firms progressed, they expanded knowledge repositories with restricted access stages. Recognising that this approach also impacts the onboarding process for the new workforce and decreases training supports the viewpoint expressed by Yeo [2].

### 4.1. Moderating Variables

To better express possible similarities and distinctions concerning the acquirement and dissemination of knowledge among individuals, both within and between organisations, through internal and external channels, startups in four different stages of maturity [75] were chosen, namely ES (early stage or surviving)—beginning of product commercialisation; LS (lifestyle stage)—operating in the regional market with a single product; GS (growth stage)—acting in the national market with more than one product; and FS (foreign stage)—internationalised.

For Carneiro [24], organisations operating in transitional economies exhibit distinct knowledge transfer patterns that vary based on their level of maturity. In this context, all startups associated their present stage of development with participation in incubation or acceleration programs. Some startups were born within a university, and others participated in processes in different states of Brazil. According to them, incubation facilitated the subsequent entry of their products/services into the national market. Four startups became involved in international programs, the latter being the only internationalised one with offices on three continents.

Validating the understanding of Acs et al. [58], there is a perception among all respondents that "connections move the world" and that incubation and acceleration programs should be guided by the networking they can provide to participating organisations, as stated: "If an incubator or accelerator, if you don't have an extensive and good network of contacts, it's useless" and "due to the incubation process, we have signed contracts ( . . . ) the networking we have is a result of these ties we create with the university and the ecosystem".

Reaffirming the findings of Bolzani [76], the industry impacted the knowledge-sharing routines assumed by entrepreneurs. Startups SA and SJ, which belong to the hard tech sector, adopted the remote culture first. SE, which has its core business linked to its patents, presented information secrecy agreements referring to intellectual property and programming codes that were formally instituted and extended to all its members. As for the number of employees, it was found that the larger the company, the greater the need to develop more robust communication channels. While SC and SG use cloud storage and WhatsApp groups as their primary tools, SA and SF have internal wikis with information flows and processes already mapped and GitHub for code storage.

### 4.2. Motivations to Share

The investigated organisations claimed to see the acquisition and sharing of information as a discriminating factor for the company's success. Based on the literature, the presented study framework organised knowledge management into several areas proposed by Clauss [59]. Each one emphasises topics divided between external and internal motivations to the individual.

### 4.2.1. External Motivations

Related to perceived reciprocity, community knowledge-sharing culture can occur intra- and inter-organisationally [77]. In this regard, it was observed that all interviewed startups dynamically engage in online communities, contributing and acquiring knowledge while fostering a culture of information sharing among employees. Moreover, information sharing among team members identified a notable absence of competition or dominance. On the contrary, interviews exhibited a sense of communalism and collaboration, aligning with existing literature that suggests an organisational climate emphasising competition between individuals hinders the development of a sharing culture [78].

The statements from the interviewees highlight that sharing information yields benefits such as gaining recognition, building credibility, and potentially establishing closer connections with investors and mentors. This understanding emphasises the significance of implementing organisational incentives and prizes to motivate people to share their knowledge [13].

Diverging from the understanding of Kansheba [78], which indicates heterogeneity as a predictor of integration and sharing difficulties, the study presented heterogeneity as a positive factor among the team. However, SD offers some communication failures among the team members due to language differences.

### 4.2.2. Internal Motivations

The interviews reveal that the integration of the team, both within and outside the work environment, plays a significant role in influencing knowledge sharing among employees. This observation supports Blair's perspective [14] that individuals tend to prioritise sharing information with peers whom they trust.

Incentive, recognition, and intra-organizational reward campaigns promote knowledge sharing [78]. This attests to the findings found in the startups interviewed that reported having credit and meritocracy as one of their pillars to encourage greater integration of knowledge. The case of the SA company is also noteworthy, where some of the employees became partners based on the performance achieved.

### 4.3. Specificities of Knowledge Sharing

### 4.3.1. Nature of Knowledge

Power equations within the investigated organisations have been shown to drive the paths of knowledge sharing dynamically but are equally dependent on social relationships between individuals to create, share, and use information and systems. They were detected in all the companies cross-examined; this usually occurs more casually than via official communication guides, and much of this procedure is determined by the startups' beliefs [52].

The hierarchical grade within an organization directly impacts the type of information shared and the level of path granted to members. Tacit knowledge is exchanged through virtual or presential daily meetings, fostering interaction between senior and junior individuals. As startups have unique characteristics, their lean structure limits the restricted ability of the founding partners, focusing on essential insights [7].

Explicit knowledge, on the other hand, is formalised by supervisors and does not demand endorsement from top-level management. Anyone within the organisation can propose this knowledge. The primary advantage reported by the surveyed firms in normalising knowledge is the reduced onboarding time for new employees and the diminished necessity for outer prime training. This is due to inner processes previously being identified, promoting organisational learning [7,52].

### 4.3.2. Sharing Type

Interorganizational knowledge transfer was observed from the vista of association relationships [79], focusing on relationships and correspondences between the knowledge provider and receiver [49]. The choice to share or acquire knowledge in and out of organisations was based on individual judgment, considering factors such as the recipient's absorption capability, the impetus to teach and learn, and internal group transfer capability, aligning with findings by Tanev [80].

While all firms displayed pro-knowledge-sharing values [43,48], with active support from managers and recognition practices, sharing knowledge was strengthened among employees who perceived themselves as specialists in the relevant subjects. Consistent with Carneiro [24], the main reasons for providing knowledge among partners and within communities of practice were skill improvement and fostering innovation.

Regarding knowledge acquisition, the individuals interviewed generally followed established rules, aiming to replicate routines and enhance their self-perceived qualifications. This finding aligns with Carneiro's perspective [24]. E-learning emerged as the most widely utilised feed for knowledge convergence among management and operational stages [46].

### 4.3.3. Share Extension

Knowledge sharing was extensively practised among affiliates of organisations, encompassing intra-team, inter-team, and inter-organizational interactions. Individuals were inspired by peer perception and self-confidence [78]. This enthusiasm was assigned to understanding systems fostered by involved and decentralised leadership styles [80], such as talent improvement programs, outer training, inner workshops, and communal team assessments.

Those who undertook training or considered themselves proficient in certain areas willingly shared their knowledge with their team and other groups. This sharing occurred through the tacit transmission of information and the availability of explicit knowledge in repositories, such as manuals and flowcharts, fostering interdisciplinary collaboration [77]. Internal relationships, shared values, and group norms were identified as drivers of knowledge sharing [59].

The surveyed startups reported minimal conflicts that hindered the knowledge-acquiring and knowledge-sharing practices [26]. Instead, they cited external influences such as public judgment, organisational status, and market fluctuations as potential influences. Only SF, which operates internationally, emphasised the significance of regulatory changes as a crucial factor impacting sharing practices [41].

All startups actively engaged in online communities of practice, led by their members and inner leaders [81]. It was noted that strong connections fostered more sensitive closeness, and social connections facilitated knowledge transfer and improved the value of exchanged information.

### 4.4. Perceived Results

Interactions between organisations were viewed as an energetic "feedback loop" formation, where knowledge exchange occurs in ever-changing backgrounds that foster learning. The feedback bonds among peers and instant supervisors and the causal ties among distinct organisational knowledge were identified as immediate influencers of firm results [25]. All interviewees recognised the intentional benefits of networking and expressed satisfaction with the balanced advantages they gained. They highlighted the positive outcomes of sharing knowledge between organisations, particularly in expanding their network of relationships. Startups emphasised that participation in incubation and acceleration programs provided access to a wide range of customers and investors. These network connections were instrumental in establishing their current market position [78].

### 4.5. Emerging Categories

The category of data security became apparent during the study as startups expressed their concern and need to safeguard the ability and critical information of their core activities. Similar to the findings of Duan [36], all startups recognised the importance of data security as a policy for market entrance and endurance. Nonetheless, it was observed that only startups at more advanced stages of maturity had implemented a contingency plan and formal agreements to avoid information escape and knowledge misadventures.

## 5. Analysis

### 5.1. Research Framework

Knowledge is described in the literature as a theoretical or practical understanding of something acquired through education or perception based on beliefs, previous experiences, values, and contextual analysis [10,52]. Volunteer participants showed an analogous but restricted comprehension of knowledge, focusing on what they can achieve with information instead of realising what they have [3]. Although literature recognises how knowledge plays a vital role in an organisation's continued survival and sustainable competitive advantage, its perception becomes more relevant in digital organisations, especially in hard tech startups.

Based on the literature review, we present a conceptual framework for studying knowledge sharing within social networks [3,63,82,83] in the context of entrepreneurship. This framework aims to facilitate the visualisation of motivations [34,84,85], distinctive characteristics [7,24,28,33,57,86–88], and the perceived outcomes [3,11,16,20,81,89] reported by the interviewees, as depicted in Figure 1. The framework proposes to represent how authors and theories about knowledge sharing are connected. Motivations represent why individuals consider sharing knowledge. The sharing process describes knowledge nature, types, and extensions that help provide understanding of how organisational culture can influence the intention to share and use knowledge management systems, how individuals' personalities interact with situational factors, and how knowledge sharing is affected. Perceived results link participants' perceptions about expected benefits to them and organisations showing how to develop personal relationships with peers, teams, and other organisations based on their adherence to knowledge share flow. Finally, the variables indicate what environmental elements could influence the knowledge path (socialisation, externalisation, internalisation, and combination [7]) through groups and organisations.

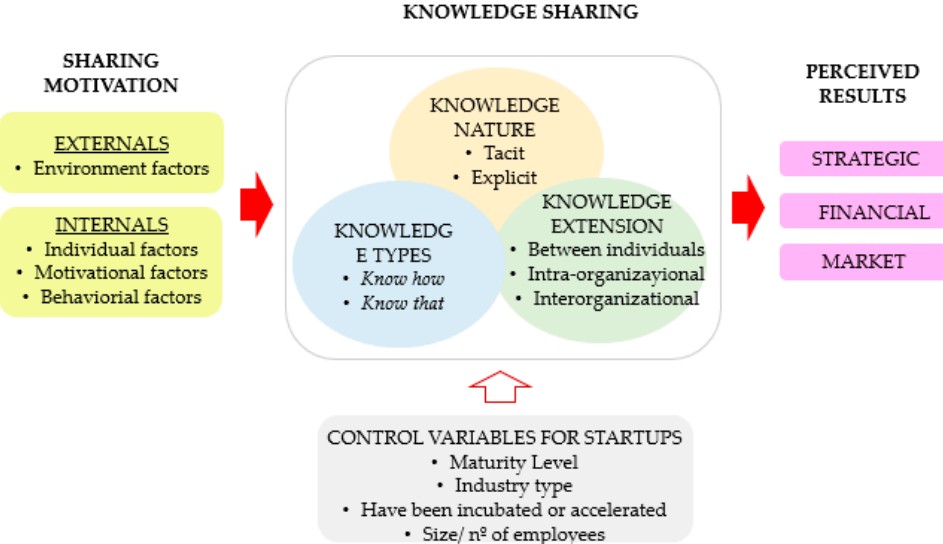

**Figure 1.** Knowledge management through entrepreneurship networks framework.

*5.2. Theoretical and Practical Contributions and Implications*

Our research is a potential contribution to developing knowledge-sharing literature and behavioural theories [24,28,33,76,87]. We aimed to be able to expand theory and practice. The tight mechanisms of knowledge sharing moulded by previous individual experiences [24,28,33,59,90] are now shaped by virtual capabilities, artificial intelligence, and cybersecurity, notably regarding intellectual property in internationalised startups. This study explores the knowledge flow in digital startups from emerging economies, particularly the motivations, means, and methods used for knowledge sharing. We use empirical analysis and extant literature [3,4,89] to make our main contributions by analysing individual and inter-organisational connections to establish, access, and disseminate knowledge channels.

The first contribution is the theoretical framework that integrates the theoretical lenses of social capital, grounded action, social exchange, and structural holes and networks [34,57,84,85], pointing to the necessity of including new behavioural and social cognitive theories. It can serve as a tool to examine the effectiveness of digital startups. The framework supports a comprehension of micro-level vital elements [91–93] to create an efficient knowledge flow to increase strategic, marketing, and financial results with intensive use of digital technologies to support scalability and product and service offerings in multiple markets.

Similar studies [58,94–98] in this field studied advanced economies and mostly with quantitative methods. Our findings reveal novel insights into how trust, self-confidence, the feedback loop, and expressed satisfaction from peers and supervisors directly affect network expansions based on reliance and truth [48]. In emerging economies, social rewards seem more significant than individual financial achievements, although they strongly concern the survival and success of these firms. Our empirical statements strengthen the arguments that the digital era, the pandemic, and the migration crisis have substantially changed most aspects of knowledge sharing [3,4,11,20].

Our findings revealed that a significant portion of knowledge sharing in startups occurs informally, driven by a sense of collectivism, trust, and recognition among individuals and organisations [99]. The critical outcomes reported were enhanced networking opportunities through incubation and acceleration programs, facilitating expanding connections [100]. Access to technical knowledge, capital, partnership market contacts, and customer acquisition was commonly mentioned [14,24,101]. This empirical study investigated various factors influencing knowledge acquisition and sharing across domains, consolidating previously explored findings.

This research also delineates social interactions to foster sustainable competitive advantages in digital startups, suggesting mixed communication channels with traditional and

digital capabilities combining cognitive, digital managerial, and multicultural capabilities. The last must consider the ability to speak different languages due to multicultural teams and markets [3,91,102,103]. Practically, the content analysis provided a comprehensive view of how the digital era, virtual relationships and open depositories, and data security [3,4,16,20,89] are embedded in environmental culture, especially in multinational teams. Practitioners, university managers, and policymakers should reflect on all the changes reported to develop novel regional or national policies to promote intrapreneurship and pedagogical evolution in management and technology curricula [4,11]. Managers should develop internal policies and mobilise resources and capabilities to enable and support better communication between members from different origins and cultures [3]. Specifically, digital entrepreneurs from internationalised startups are recognised as the top-level knowledgeable person in the firm, enabling connections among teams and outer organisations to share managerial practices, close market follow-up, and technological advances.

In terms of methodology, this study brings innovation by qualitatively examining startups operating in emerging economies with varying degrees of maturity and across different industries. It considers sharing intent and self-reported behaviours from different perspectives and hierarchical levels to address issues preventing the undue transfer of knowledge to competition and information security. We unearthed that participants from digital startups demonstrated that although they had participated in incubation and business acceleration programmes linked to universities, they needed to assign more importance to formal channels of communication and knowledge acquisition [13,55]. They considered self-learning and informal education acquired in communities of practice to be more valuable than academic qualifications to feel like experts on a subject. Virtual relationships with other organisations directly influence the individual motivation to share knowledge and the recipient's choice based on the perception of their absorptive capacity of the shared content [91,92].

### 5.3. Limitations and Future Paths for Research

The extent to which our findings can be applied broadly is constrained in various ways from reaching generalisability. To illustrate, we utilised focus groups with in-depth interviews and qualitative exploratory research approaches within a particular emergent economy. Subsequent investigations could consider employing hybrid methodologies and longitudinal research designs within the same or alternative geographical regions, business communities, or emerging economies as BRICS with different cultures. Russia, India, China, South Africa, and other countries' research can be included to explore whether our findings are unique to this specific economy. Future research could also encompass a more varied participant sample, encompassing individuals from diverse religious backgrounds, ethnic origins, genders, and sectors.

Additionally, an intriguing direction for future research would be to explore the impact of digitalisation and the culture of remote first anticipated by the pandemic, the migration crisis, and war. Ultimately, the consciousness of knowledge-sharing flow (creation, acquisition, transfer, exchange) is inducted by environmental culture, especially from relations between employees and supervisors and shared value systems. New research adding other stakeholders or theoretical lenses could show perceived results and behaviour relatively differently.

Ultimately, as all participants are around the Y and Z generations and all digital natives, new studies to analyse how artificial characters and intelligence in cyberspace should affect personal and virtual connections and motivations for knowledge sharing are welcome. In addition, we have not directly addressed the effects of the COVID-19 pandemic, migration crisis, and war, which seem to have radically changed the course of international business and accelerated the narrowing of the generation gap, notably the alpha generation (born after 2010). New studies exploring more mixed methods and ethnography should provide novel helpful insights. None of the public sector start-ups was included, which affected the perception of variation in organisational norms and culture.

## 6. Conclusions

Based on the research found, this empirical study shed light on how motivations, connections, mechanisms to share, and communication channels influence knowledge flow through digital startups in emerging economies and contribute to comprehending how startup members learn or share knowledge. Our research emphasises positive experiences in digital businesses fostering virtual practices to support sustainable competitive advantages via open innovation and collaborative processes. Our findings show why individuals consider sharing knowledge, how beliefs and culture influence inner intentions, how they interact with situational conditions, and how to develop personal relationships with peers, teams, and other organisations.

The content analysis yielded insights into the responses to the research questions. Individuals desire to share knowledge for various reasons, including seeking recognition, establishing credibility, and forging connections with investors and mentors. A communal and collaborative atmosphere can foster the exchange of information among employees. At the same time, a meritocratic culture promotes greater integration, influencing the intention to share and utilise knowledge management systems. The decision to acquire knowledge within and outside organisations is influenced by individual judgment, considering the recipients' ability to absorb knowledge and motivation to teach and learn. Sharing knowledge is reinforced when employees perceive themselves as specialists in relevant subjects. Incentives and intra-organisational reward campaigns, talent improvement programs, external training sessions, workshops, and collaborative team assessments can cultivate personal relationships among peers, teams, and other organisations.

The study sheds light on cultural and behavioural differences in digital startups born in emerging economies. Changes in the digital age, the pandemic, and the migration crisis support some of these differences. Another portion points to macroeconomic factors of these economies, such as scarcity of resources and structural and institutional gaps. In other words, this denotes that studies in advanced economies cannot be generalised to emerging economies given the divergence of digital managerial, multicultural, and cognitive capabilities. Participating in incubation and acceleration programs linked to universities for the entire sample was fundamental to ensuring access to markets and investors via networking. The developed sense of belonging stimulated knowledge sharing in virtual communities of practices in tacit and explicit forms. There was a shared belief among respondents that connections play a crucial role, and access to networks was the most valuable benefit provided by institutions. Incentives, appreciation, and internal prize initiatives were implemented to promote knowledge sharing, with a merit-based culture as a foundation to encourage greater integration of knowledge. At last, we proposed a framework that could serve as a tool to comprehend the path of knowledge sharing in digital startups based on qualitative methods.

**Author Contributions:** Conceptualisation, R.D.d.A. and P.G.P.; methodology, R.D.d.A. and P.G.P.; software, R.D.d.A., M.D.M.P. and T.L.D.P.; formal analysis, R.D.d.A.; investigation, R.D.d.A.; data curation, R.D.d.A.; writing—original draft preparation, R.D.d.A.; writing—review and editing, P.G.P.; supervision, P.G.P.; funding acquisition, R.D.d.A., P.G.P. and M.D.M.P. All authors have read and agreed to the published version of the manuscript.

**Funding:** This research was funded by the NECE—Research Center in Business Sciences funded by the Multiannual Funding Program of R&D Centers of FCT—Fundação para a Ciência e Tecnologia, Portugal, under grant UIDB/04630/2020. This research is also supported by a grant sponsored by FCT, under grant 2022.14694.BD.

**Institutional Review Board Statement:** Not applicable.

**Informed Consent Statement:** Informed consent was obtained from all subjects involved in the study.

**Data Availability Statement:** Not applicable.

**Acknowledgments:** We wish to convey our appreciation to all the individuals involved in this research, who graciously and precisely responded to our inquiries patiently.

**Conflicts of Interest:** The authors declare no conflict of interest.

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
