# Peer review of "Unleashing Knowledge Sharing in Emerging Economy Startups: A Multilevel Analysis"

_sustainability, doi:10.3390/su151310338_

Round 1

Reviewer 1 Report

Revise Abstract as in the current form is not clear what approach you followed and what the results are. Clearly indicate them.
Abstract – you wrote “ this 9 article aims to analyse the similarities and distinctions regarding the creation, acquisition, and shar-10 ing of knowledge in startups.” whereas the paper focus on knowledge sharing and the RQs are about KS exclusively.

Justify this decision “sharing knowledge or information adopted in this research as synonyms”. Also, make a reliable lit.review on information and knowledge and decide which of them you should use. Do not simplify the theory which is huge and has a lot of linked directions.

Due to the simplification mentioned earlier, Figures 2 and 3 contain numerous variables and concepts without sufficient background or explanation. Therefore, it is strongly recommended to conduct a more extensive literature review to address this issue.

 3.2. Research Methods is explained too briefly. It would be helpful to understand the precise decision-making process that guided your research. Consider presenting it in a step-by-step manner.

Results and Analysis.

The first sentence as an example “As for knowledge management in startups, all had some mechanism for collecting, 191 storing, and distributing explicit knowledge.” In this sentence you have mentioned about knowledge management, K.storing, K.distributing, and explicit knowledge. Nothing about them was developed so far.

The entire "Results" chapter should adhere to a logical order and be referenced clearly with respect to the aforementioned categories and framework.

What about research questions? Where are you addressing these questions?

Author Response

Dear reviewers,
In principle, I thank you for all your contributions and requests for revisions. All of them were met, and the article was entirely rewritten. Many of the modifications pointed in the same direction, so I took the liberty of reporting them together as comments below by section.

ABSTRACT:
It has been entirely rewritten to improve the approach and report the main results. It also added the meaning of the topic, motivation for the article, research gap, main objective (partial ones are in the introduction and theoretical framework), material and methods, findings, and prospects.
LITERATURE REVIEW:
All the concepts of sharing, exchange transfer, knowledge creation, and acquisition were added to demonstrate the differences and what the authors considered similarities and differences. The same was done for knowledge, data, and information. This section has been extended, and we have added new, more relevant, and more recent studies, including papers suggested by some reviewers. The study was placed in a broad context, its importance highlighted, citations updated, the research gap specified, and the main objective. At the end of each session, the specific goals that served as a premise for the investigation were added/linked, connecting them with previous literature. There is no need to talk about hypotheses as this is not a quantitative study.
METHODS:
It was expanded, specifying the decision-making process/ criteria that guided the research. The step-by-step sample selection, eligibility criteria and content analysis were presented. The data collection period was also specified, as well as the use of focus groups. The interview cut-off criterion of saturation was explained and detailed. Since this was not a quantitative study, there was no sample calculation—the non-generalization of data inherent to qualitative studies, as described in the analysis and limitations section. Observational data were cut as requested.
RESULTS AND ANALYSIS/DISCUSSION:
These were in the same section and were split as requested. Results were reported according to the interview content categorisation tree for easy understanding. All are connected to existing literature and demonstrate new findings.
The analysis section was divided into three sub-sections: research framework, theoretical and practical contributions and implications, and limitations and future paths for research.
Here it was explained how the objectives were met, it was discussed which part of the findings followed the existing literature, and discoveries were reported contrasting with the comprehensive theoretical framework. The results that did not connect with the theoretical framework were linked with new recent papers.
CONCLUSION:
We followed the suggestion to narrow it down and used the text suggestion to make some changes. This section remained with only two paragraphs. Since the limitations, theoretical and practical contributions, implications, and future lines of research were previously reported.

FIGURES:

Figures 1 and 2, which appear in the method and deal with the sample characterisation and content categorisation tree, were detailed step-by-step in the text. Manual inputs in NVivo generated Figure 2 but cannot be exported as it is a column (nodes, attributes, and relationships). I tried to turn the screen screenshot into a figure, but it looks terrible; that’s why it's in Excel. From NVivo, I used the reports. After classifying all the lines, it is possible to generate reports for each node and attribute, and this greatly facilitates the interpretation of the lines by isolating and grouping the main ones.

Figure 3, out of context, was transferred to the discussion and detailed throughout the text as it relates to the theories used and content analysis. My version of NVivo allows you to do something along these lines and export, but it has limitations on colours, formats, and fonts. It is not a figure that the software automatically generates. So it would be the same result, only ugly. An exciting output would be the connection tree, but the figure is enormous, the letters are unreadable, and the software does not entirely pull the highlighted sections. In practical terms, for this study, the reports that grouped the already selected and categorised speeches were interesting to us. Part of these speeches was reported in the results section.

OTHERS:

*Language proofreading was done in Grammarly software and by a native speaker.* The term sustainable competitive advantage has been defined extensively in all article sections.

Reviewer 2 Report

Dear authors,

1. It is necessary to specify the results in the Abstract.

2. In accordance with the stated goal of the article in the literature review, it would be appropriate to indicate the authors and the directions of their research regarding startups in different industries that are currently developing.

 3. It is necessary to outline the criteria based on which the selection of one or more processes, activities, or individuals over a limited period was conducted on which a multiple study

4. It would be desirable to indicate the expected practical implications of your conducted research.

5. The article lacks argumentation regarding the selection of the focus group for the conducted research.

6. The conclusions have a descriptive nature and are a continuation of revealing the essence of the conducted research. It is advisable to specify the results of the conducted research in the conclusions, indicating the identified areas for improvement in the respective direction.

7. The conclusions need to be finalized taking into account the results obtained.

Author Response

(The authors gave the same response as above.)

Reviewer 3 Report

Dear/s Author/s,

Re: Manuscript “Unleashing Knowledge Sharing in Emerging Economy Startups: A Multilevel Analysis

Reviewer’s report:

The aim of this article is to analyze the similarities and differences in the creation, acquisition and sharing of knowledge in startups. This type of companies, so relevant nowadays, depend on knowledge and it is convenient to know how they obtain and use this knowledge. From the introduction and the literature review it is not clear what the research gap is. The methodology provides references, but does not justify why the qualitative method is used. The companies studied are not justified either. In order to be in-depth interviews, they may be scarce. The authors can use and justify the saturation of data produced with these interviews. The sections on results and analysis should be separated into results and discussion. The latter is where the results should be contrasted with the previous theoretical framework. The conclusions are excessively long. A short paragraph stating whether the objectives have been met is sufficient. The rest should be transferred to the discussion section.

Best regards

Author Response

(The authors gave the same response as above.)

Reviewer 4 Report

The reviewed paper deals with an interesting topic. The quality of this paper must be improved. The authors are required:

- to write a structured abstract (the meaning of the topic, motivation to write a paper-or make a research, the main aim, partial aims, shortly: material and used methods; the main findings and outlook to the future)

- improve the theoretical background. You can use inspirations and new ideas from: for example this references: 

Mura, L., Zsigmond, T., & Machová, R. 2021. The effects of emotional intelligence and ethics of SME employees on knowledge sharing in Central-European countries. Oeconomia Copernicana12(4), 907–934. https://doi.org/10.24136/oc.2021.030

Mura, L., Krchova, H., & Chovanova Supekova, S. (2021). Managing the development of innovative and start-up forms of businesses and verification of INMARK concept. 1st Edition. Szczecin: Centre of Sociological Research, 131 p. ISBN 978-83-963452-2-6.  https://doi.org/10.14254/978-83-963452-2-6/2021

- what figure is an output from NVivo software? In the paper text there are only standard tables and picture creating in word

- to write a structured conclusions (the main aim, the main problems and limits in research, the main findings with arguments, background for the next research in this topic.

The quality of the used English can be improved. 

Author Response

(The authors gave the same response as above.)

Reviewer 5 Report

Thanks to the authors and editors of Sustainability for the opportunity to read and review the manuscript of ‘Unleashing Knowledge Sharing in Emerging Economy’ Startups: A Multilevel Analysis’ Sustainability-2426125.

The results of the study indicate the main reasons perceived by individuals to share knowledge, how their characteristics interact with situational factors, and how they reflect on the development of social relationships. They also reveal that, despite the concern with information security, confidentiality agreements are standard when there is a perception of the link between intellectual property and the organisation's core business.”

The paper offers some interesting insights and adequate methods, and the topic seems interesting, but serious issues are detected, and various corrections are needed to upgrade its current level. For instance, in several areas, including its grounding in the abstract, introduction, sample design, implications, etc.

Serious points:

Serious points:

·        It is not clear how the article relates to sustainability and the profile of the journal. What does “sustainable competitive advantage in a dynamic economy” mean?

·        The authors should consider reframing the paper and submitting it to an appropriate English language for readers.

Additional amendments:

·        Abstract. Place the research question addressed in a broad context and highlight the purpose of the study; briefly describe (not order) the main methods or treatments applied; summarize the article's main findings, and indicate the main conclusions or interpretations.

·        The introduction part needs to be adequate. The introduction should briefly place the study in a broad context and highlight its importance. It should define the purpose of the work and its significance. The current state of the research field should be carefully reviewed, and key publications cited. A research gap should also be emphasized in why the topic is selected. In the introduction, answer two main questions: "Why was this particular study needed to fill this gap in current scientific knowledge?" Describe the rationale in the context of what is known and not in this research area. The problem statement based on the previous results and the novelty of this study should also be declared here.

·        The study's theoretical background should be provided with an explicit statement of the hypothesis being addressed concerning literature comparisons, outcomes, study design, etc. Hypotheses are crucial to appreciate how it relates to the previous theoretical and empirical literature.

·        Data selection should be explained more. To maintain the integrity, transparency and reproducibility of research records, authors must make their experimental and research data openly available either by depositing them into repositories or by publishing the files as supplementary information in this journal. The data design section does not seem appropriate for describing the investigation of data collection. Refer carefully to the literature on how and why this case study and the materials, and data design are selected.

·        If authors collect any data, observational, surveys or secondary data, it needs to decide what data to collect and from whom. The point of sample selection is to generalize the findings to the whole population, which means that the sample must be adequate. There are two options to choose from (1) representative of the population. In other words, the sample should include subgroups in proportion to the total population. It should not exclude certain groups because of the sampling method, the design or the choice of respondents. (2) large enough to provide sufficient information to avoid errors. It need not be a fixed proportion of the population, but it should at least be of a specific size so that you know that your answers are likely to be broadly correct. Currently, it is unclear what criteria and grouping were used to select the companies to analyze. Hence, sample seems to be not representative, and the results cannot be generalized for other regions and studies and these issues should be discussed as limitations.

·        Recent study has also stressed the potential problem of common method bias, which describes the measurement error compounded by the sociability of respondents who want to provide positive answers (i.e. perceptions) (Chang, v. Witteloostuijn and Eden, 2010). Thus, raising potential common method variance as false internal consistency might be present in the same data and provided results.

·        More sophisticated qualitative methodologies are expected for such high-ranked MDPI journals.

·        The implication/conclusion chapter should go beyond the interpretation of the results. For instance: - The main objective of this study was to examine .... effects of ….. on the........, to shed light on novel research perspectives on ..... - A .................. analysis was used to calculate impacts between ..... while also taking ....... into account ….. - The advantage of the research model is that .......  - Contrary to previous approaches, we consider -----. We found that ...... . Namely: (a), (b), (c) ....  - The methodological/theoretical implication is that ........ The findings are also important for policymakers ………. – Limitations - Future research direction, etc.

.

Author Response

(The authors gave the same response as above.)

Round 2

Reviewer 2 Report

Dear authors,

A noticeable significant improvement to your paper, but unfortunately the following remarks remain:

1. The abstract should be improved from line 18, namely: "The sense of collectivity, cooperation and trust in partners...." to line 24 "support startups and integrate entrepreneurial ecosystems". Authors should show the results and conclusions of the study, not general phrases.

2. Conclusions should also be reinforced by specific research results.

Author Response

Dear reviewer,
Once again, we thank you for your significant contributions. As requested, changes have been made in the abstract and conclusion that can be tracked in the markups.
More specific conclusions have replaced lines 18-24 of the abstract, and the particular results have been reinforced with a second paragraph in the decision.
These were the only changes in the text of the article since the other reviewers did not request further changes.

ABSTRACT (new replacement): "Individuals desire to share knowledge for recognition, establish credibility, and connect with investors and mentors. A communal and collaborative atmosphere can foster the exchange of information among employees, influencing the intention to share knowledge. Knowledge sharing is reinforced when employees perceive them as specialists. Incentives and intra-organisational reward campaigns, talent improvement programs, external training sessions, workshops, and collaborative team assessments can cultivate personal relationships. A theoretical framework has been proposed that can examine digital startups' effectiveness on micro-level elements. In emerging economies, social rewards seem to be personally more critical than financial achievements. Our empirical statements reinforce the arguments that the digital age, pandemic and migration crisis have substantially changed most aspects of knowledge sharing."

CONCLUSION (second paragraph added): "The content analysis yielded insights into the responses to the research questions. Individuals desire to share knowledge for various reasons, including seeking recognition, establishing credibility, and forging connections with investors and mentors. A communal and collaborative atmosphere can foster the exchange of information among employees. At the same time, a meritocratic culture promotes greater integration, influencing the intention to share and utilise knowledge management systems. The decision to acquire knowledge within and outside organisations is influenced by individual judgment, considering the recipients' ability to absorb knowledge and motivation to teach and learn. Sharing knowledge is reinforced when employees perceive themselves as specialists in relevant subjects. Incentives and intra-organisational reward campaigns, talent improvement programs, external training sessions, workshops, and collaborative team assessments can cultivate personal relationships among peers, teams, and other organisations."
At last, the language was proofread again by Grammarly software on five computers to optimise the algorithm and a native speaker.

Thank you so much for your attention and participation.

Reviewer 3 Report

Dear/s Author/s,

Re: Manuscript “Unleashing Knowledge Sharing in Emerging Economy Startups: A Multilevel Analysis”

Reviewer’s report:

Once the great news contributed by the authors has been observed, my opinion is to recommend the publication of this paper.

Best regards

Author Response

Thank you! :)

Reviewer 5 Report

The study has been significantly strengthened and fits into the profile of the journal. Responses to the improvements are acceptable. 

I am not qualified to assess the final quality of English in this paper. MDPI language support can be advised.

Author Response

The file was re-visited using the Grammarly software on five different computers (to ensure the best use of the algorithm). On all of them, the errors were corrected and zeroed. Finally, one native speaker re-evaluated and did not make any corrections.

We would like to thank you again for all your contributions.